# Convergence of Stochastic Gradient Langevin Dynamics in the Lazy Training Regime

**Noah Oberweis**                                          *oberweis@mathc.rwth-aachen.de*
*Department of Mathematics*
*RWTH Aachen University*

**Semih Cayci**                                               *cayci@mathc.rwth-aachen.de*
*Department of Mathematics*
*RWTH Aachen University*

**Reviewed on OpenReview:** *https://openreview.net/forum?id=3sO37DKmLo*

## Abstract

Continuous-time models provide important insights into the training dynamics of optimization algorithms in deep learning. In this work, we establish a non-asymptotic convergence analysis of stochastic gradient Langevin dynamics (SGLD), which is an Itô stochastic differential equation (SDE) approximation of stochastic gradient descent in continuous time, in the lazy training regime. We show that, under regularity conditions on the Hessian of the loss function, SGLD with multiplicative and state-dependent noise (i) yields a non-degenerate kernel throughout the training process with high probability, and (ii) achieves exponential convergence to the empirical risk minimizer in expectation, and we establish finite-time and finite-width bounds on the optimality gap. We corroborate our theoretical findings with numerical examples in the regression setting.

## 1 Introduction

Stochastic gradient descent (SGD) has been the main workhorse of deep learning due to its low computational complexity and effectiveness. The existing theoretical analyses of first-order optimization methods in deep learning predominantly study the deterministic (full-batch) gradient descent, which neglects the impact of inherent stochasticity of SGD in training neural networks. Although there exist recent works that investigate stochastic dynamics (see, for example, Lugosi & Nualart (2024)), explicit finite-time convergence rates for the stochastic first-order methods under realistic and interpretable conditions still remain elusive, which motivates our work.

In this work, we establish theoretical guarantees for stochastic gradient Langevin dynamics (SGLD), a continuous-time approximation of SGD, in the lazy training regime. One important application of our theoretical analysis is finite-time and finite-width bounds on the training loss for deep feedforward neural networks trained by SGLD in a supervised learning setting. The main contributions of our work include the following:

- **Exponential convergence rates in expectation:** By leveraging tools from stochastic calculus, we derive a stochastic Grönwall-type inequality for SDEs in the lazy training regime, which establishes an exponential decay rate for the expected optimality gap as long as the parameters are within a certain neighborhood of the initialization. To that end, we explicitly characterize the impact of the curvature of the loss function in the parameter space and the output scaling factor on the effective noise during the training process. The resulting scaling law describes sufficient conditions on the curvature of a global minimum to prevent SGLD from escaping it, which is one of the key outcomes of our analysis.

- **High probability bounds on the conservation of the lazy training regime:** A key quantity in the lazy training analysis is the first-exit time at which the parameters leave a certain neighborhood of their random initialization, which lower bounds the duration of descent throughout the optimization steps. Leveraging our stochastic error analysis, we derive high-probability bounds on this first-exit time, and show that, for sufficiently large output scaling, the parameters remain in this neighborhood indefinitely with high probability, thereby completing the convergence argument in the stochastic setting.

- **Applications in deep learning:** We apply our theoretical analysis to derive explicit non-asymptotic (finite-time and finite-width) bounds on training loss for deep neural networks trained by SGLD in the lazy training regime. To the best of our knowledge, this is the first analysis of stochastic gradient Langevin dynamics in this regime.

Our work builds on and extends the deterministic lazy training analysis in Chizat et al. (2020), which analyzes the convergence of gradient flow for overparameterized neural networks in the lazy training (or the so-called kernel) regime. In Chizat et al. (2020), it was demonstrated that the lazy training phenomenon arises from an appropriately chosen output scaling factor, which ensures that the neural network parameters remain in a small enough neighborhood of their random initialization. We note that Chizat et al. (2020) considers deterministic (full-batch) gradient flow in the lazy training regime, and does not address the stochasticity that stems from stochastic subsampling in SGD, which is crucial for learning with large datasets. In this work, we extend this framework by analyzing a stochastic continuous-time approximation of the SGD.

## 1.1 Related Work

**Convergence of overparameterized neural networks in the kernel regime.** Overparameterized neural networks have been shown to achieve interpolation, which implies zero training error when they are trained with first-order methods despite the highly non-convex optimization landscape. At the same time, they achieve impressive generalization performance even with noisy data (Belkin et al., 2019; Bartlett et al., 2020; Zhang et al., 2021). Additionally, works such as Oymak & Soltanolkotabi (2019) have demonstrated that the overparameterized regime can be reached with fewer parameters than previously suggested by the theory. The theoretical explanation of this phenomenon has been a focal point of interest. It was shown in Jacot et al. (2020); Du et al. (2019b); Li & Liang (2018) that randomly-initialized overparameterized neural networks, trained with gradient descent, achieve global optimality without moving away from their random initialization, which is known as the lazy training or kernel regime. The main inspiration for this work is Chizat et al. (2020), where lazy training is first analyzed as a consequence of an artificial scaling factor. The existing works in the lazy training regime predominantly analyze overparameterized neural networks trained with (full-batch) gradient descent, and they do not extend to stochastic gradient descent, which is the core focus of our work. Additionally, it is essential to point out that, contrary to other works, our primary goal is not the analysis of linearized dynamics defined by $\bar{h}(\omega) := h(\omega_0) + Dh(\omega_0)(\omega - \omega_0)$. It can be shown that gradient flow stays arbitrarily close to the linearized dynamics for sufficiently large $\alpha$. We will use very similar techniques to prove an exponential convergence rate of the empirical error for a sufficiently large $\alpha$ without looking into the case where $\alpha \to \infty$.

**Convergence of stochastic gradient descent for neural networks.** Dynamics of stochastic gradient descent have been investigated in Lugosi & Nualart (2024) for deep linear neural networks, which shows that a global optimum is achieved under some regularity assumptions on the loss landscape with a convergence rate that has an implicit dependency on the problem parameters. In our work, inspired by the analysis in Lugosi & Nualart (2024), we analyze the stochastic gradient dynamics for training non-linear neural networks, and establish exponential convergence to the global optimum with an explicit and interpretable convergence rate under easily verifiable regularity assumptions.

**Modeling of discrete training algorithms by SDEs:** Continuous-time modeling of discrete-time stochastic gradient descent by using SDEs has recently been investigated in many works, as the SDE model provides essential insights into the discrete stochastic gradient descent training algorithm. In this work, we consider an SDE-model to approximate stochastic gradient descent dynamics, inspired by Li et al. (2017), which proved the viability of using stochastic gradient Langevin dynamics to model mini-batch stochastic

gradient descent. Under certain assumptions, Li et al. (2017) shows that mini-batch stochastic gradient descent is a weak order-one approximation of the stochastic gradient Langevin dynamics equation 1 (see Remark 1).

Our work is closely related to Chizat et al. (2020) and Lugosi & Nualart (2024). However, there are many substantial differences between prior work and our contributions. For instance, Chizat et al. (2020) does not consider the stochastic term needed to model stochastic learning algorithms. Compared to Lugosi & Nualart (2024), we investigate the dynamics in the lazy training regime, which allows us to formulate less restrictive and easily verifiable assumptions in a machine learning context. An in-depth discussion on the differences of our contributions and prior works is presented in Appendix A.

We limit our results to continuous models (as defined in the following section), although training typically employs stochastic gradient descent or other iterative methods in most practical applications. In addition to the multiple technical advantages arising from the continuous approximation of real (discrete) dynamics, such as the possibility of utilizing many tools from Itô calculus and the independence of the step size of the algorithm, this choice has yielded significant insights that apply to discrete models. One such instance is presented in Malladi et al. (2024), where the authors derive a continuous model for the adaptive gradient algorithms, which is then used to motivate the optimality of the square root scaling rule for adjusting hyperparameters as batch sizes increase.

## 1.2 Notation

*Id* denotes the identity matrix. $A \preceq B$ indicates that $B - A$ is positive-definite. $\|x\|_2 = \sqrt{x^T x}$ denotes the 2-norm of $x \in \mathbb{R}$. We use $\lambda_{min}(A)$ for the smallest eigenvalue of a Matrix $A$. For a positive-definite matrix $A$, we define the vector norm $\|x\|_A^2 := x^T A x$. Additionally, $\|A\|_F$ is the Frobenius norm of $A$ and $B_r(x_0)$ is the open ball of radius $r$ around the $x_0$, in the Euclidean norm. We denote the modulus of Lipschitz continuity of a function $f$ as $\text{Lip}(f)$. The notation $Dh$ denotes the Jacobian of $h$. Similarly, $\nabla h$ represents the gradient of $h$ and $\nabla^2 h$ represents its Hessian. For a function $f : \mathbb{R}^n \to \mathbb{R}$, it holds that $\nabla^T f = Df$. We add an index to $\nabla$ to indicate, that the gradient is only taken with respect to one of the arguments. $\nabla_h$ can be seen as the gradient in the Hilbert space $\mathbb{F}$. However, this can be replaced by the standard gradient with respect to the values that $h \in \mathbb{F}$ takes on the finite set $\{x_i\}_{i=1}^n$. Lastly, $\mathbb{1}_A$ denotes the characteristic function, which is equal to 1 if the logical expression $A$ holds true and 0 otherwise.

## 2 Problem Setup and Stochastic Gradient Langevin Dynamics

### 2.1 Problem Setting

In this work, we consider a smooth parametric predictor $h : \mathbb{R}^p \to \mathbb{F}$, where the parameter space is $\mathbb{R}^p$ and the output space is a Hilbert space $\mathbb{F}$ (equipped with the norm $\| \cdot \|_{\mathbb{F}}$). In addition, we assume that the empirical risk $R : \mathbb{F} \to \mathbb{R}_+$ and the loss function $\ell : \mathbb{R}^d \times \mathbb{F} \to \mathbb{R}$ are smooth. Both functions are related by $\mathbb{E}_x[\ell(x, h(\omega))] = R(h(\omega))$, where $\mathbb{E}_x$ denotes the expected value with regards to some probability distribution $P$ from which the data points $x \sim P$ are sampled independently. The main objective in this paper is to solve the following problem:

$$\text{Find } \omega^\star \text{ s.t. } R(h(\omega^\star)) < \epsilon,$$

for some $\epsilon > 0$. Empirical risk minimization in supervised learning is a special case of the formulation above, where $P$ is a discrete probability measure on a given set of data points $\{(x_i, y_i)\}_{i=1}^n$ and an unknown ground truth $h^\star \in \mathbb{F}$, that connects $x_i$ and $y_i$ by $h^\star(x_i) = y_i$. In empirical risk minimization, a popular choice for the loss function is given by $\ell(x, h) = (h(x) - y)^2$, where $y = h^\star(x)$. For further details about the supervised learning formulation, see Section 4, where we apply the results to train neural networks with smooth activations in a supervised learning setting.

## 2.2 Lazy Training as Stochastic Langevin Dynamics

We consider the scaled stochastic gradient Langevin dynamics defined by the Itô SDE

$$d\omega_t = -\frac{1}{\alpha}D^T h(\omega_t)\nabla_h R(\alpha h(\omega_t))dt + \frac{\sqrt{\eta_\alpha}}{\alpha}(\Sigma_\alpha(\omega_t))^{\frac{1}{2}}dW_t, \tag{1}$$

where $W_t$ is a standard Brownian motion in $\mathbb{R}^p$, $\Sigma_\alpha(\omega)$ is the covariance matrix of $\nabla_h\ell(\cdot, \alpha h(\omega))$ and $\eta_\alpha, \alpha \in \mathbb{R}_+$. We adopt a random initialization $\omega_0$ such that $h(\omega_0) = 0$ holds almost surely. For neural networks, this can be achieved by using the symmetric initialization scheme (Bai & Lee, 2020; Chizat et al., 2020; Cayci & Eryilmaz, 2025).

Here, $\sigma_\alpha(\omega) := (\Sigma_\alpha(\omega))^{\frac{1}{2}}$ denotes the matrix square root $\sigma_\alpha^T(\omega)\sigma_\alpha(\omega) = \Sigma_\alpha(\omega)$, which is well-defined since $\Sigma_\alpha(\omega)$ is symmetric positive definite. The noise scaling factor $\eta_\alpha$ may depend on the output scaling factor $\alpha$ in certain cases (see Appendix B).

**Remark 1.** *For the empirical risk minimization problem where the risk function is defined by $R(h(\omega)) = \frac{1}{m}\sum_{i=1}^m \ell(x_i, h(\omega))$, the SDE in equation 1 is a continuous-time approximation of the stochastic gradient descent (SGD)*

$$\omega_{k+1} = \omega_k - \frac{\eta_\alpha}{\alpha}D^T h(\omega_k)\nabla_h R(\alpha h(\omega_k)) + \frac{\sqrt{\eta_\alpha}}{\alpha}V_k, \tag{2}$$

*where $V_k := \sqrt{\eta_\alpha}(D^T h(\omega_k)\nabla_h R(\alpha h(\omega_k)) - D^T h(\omega_k)\nabla_h l(x, \alpha h(\omega_k)))$ and $x \sim \mathrm{Unif}\{x_1, \ldots, x_n\}$ is a sample from the distribution on the input space. In that case, $V_k|x_k$ is a random vector with mean 0 and covariance matrix $\eta_\alpha\Sigma_\alpha(x_k)$. In Li et al. (2017) it is shown that the stochastic continuous-time model that we consider in equation 2 is an order-one weak approximation of the SDE equation 1. There are other stochastic models to approximate SGD in continuous time (Li et al., 2017; Simsekli et al., 2019; Gurbuzbalaban et al., 2021).*

# 3 Convergence of Stochastic Gradient Langevin Dynamics in the Lazy Training Regime

In this section, we show the convergence in expectation of the loss function under suitable regularity assumptions. Contrary to other SDE models, equation 1 contains a noise term that vanishes at a minimizer. Due to the introduced regularity of the loss $\ell$ in the following Assumption 1 and curvature constraint from Assumption 2, we can effectively keep the parameters from escaping a minimum.

**Assumption 1.** *We make the following assumptions.*

- *Regularity of the loss function: $\ell : \mathbb{R}^n \times \mathbb{F} \to \mathbb{R}$ is Lipschitz-smooth*

$$||\nabla_h\ell(x, h_1) - \nabla_h\ell(x, h_2)||_2 < \mathrm{Lip}(\nabla_h\ell)||h_1 - h_2||_\mathbb{F},$$

  *and m-strongly convex*

$$(\nabla_h\ell(x, h_1) - \nabla_h\ell(x, h_2))^T(h_1 - h_2) \geq m||h_1 - h_2||_\mathbb{F}.$$

- *Regularity of the model: The derivative of the model $h : \mathbb{R}^p \to \mathbb{F}$ is $\mathrm{Lip}(Dh)$-Lipschitz continuous, and the neural tangent kernel at initialization is strictly positive definite $\lambda_{\min}(Dh(w_0)D^\top h(w_0)) > 0$.*

Assumption 1 holds for many loss functions, with the most common example being the squared error (see Appendix B). The smoothness assumption on $h$ holds for shallow neural networks (see Corollary 1). For deep neural networks, we will present an alternative that circumvents the smoothness of $h$ (see Corollary 2 and Appendix C). The empirical neural tangent kernel is strictly positive definite for overparameterized neural networks with high probability over the random initialization under reasonable data regularity conditions (Du et al., 2019b;a; Banerjee et al., 2023).

The following assumption on the Hessian of the risk function is critical for the analysis in the stochastic setting.

**Assumption 2.** *We assume that $Tr\left(\frac{\sigma_\alpha(\omega)^T \nabla^2_\omega R(\alpha h(\omega))\sigma_\alpha(\omega)}{R(\alpha h(\omega))}\right) \leq \frac{2\alpha^2 \lambda^2 m}{\eta_\alpha}$, for all $\omega \in B_r(\omega_0)$, where $r$ is the radius that will be discussed later in this section (before Theorem 1).*

Assumption 2 automatically holds for sufficiently overparametrized shallow and deep neural networks under realistic assumptions. In Appendix B and C, we provide the proof for shallow and deep neural networks with smooth activation functions.

**Assumption 3.** *Assume that the gradient and the expectation of the loss interchange, i.e., $\nabla_h R(h) = \mathbb{E}[\nabla_h \ell(x, h)]$.*

For the empirical risk minimization in supervised learning, Assumption 3 automatically holds due to the uniform sampling distribution over a finite training set.

**Lemma 1** (Regularity of $R$). *Under Assumption 1 and Assumption 3, $R$ is $\mathrm{Lip}(\nabla_h \ell)$-smooth and $m$-strongly convex.*

*Proof of Lemma 1.* Using the convexity from Assumption 1, we have

$$\|\nabla_h R(h_1) - \nabla_h R(h_2)\| = \|\mathbb{E}_x[\nabla_h \ell(x, h_1) - \nabla_h \ell(x, h_2)]\|_2$$
$$\leq \mathbb{E}_x[\|\nabla_h \ell(x, h_1) - \nabla_h \ell(x, h_2))\|_2]$$
$$\leq \mathbb{E}_x[\mathrm{Lip}(\nabla_h \ell)\|h_1 - h_2\|_\mathbb{F}]$$
$$= \mathrm{Lip}(\nabla_h \ell)\|h_1 - h_2\|_\mathbb{F}.$$

Similarly, we can use the $m$-strong convexity from Assumption 1 to obtain

$$(\nabla_h R(h_1) - \nabla_h R(h_2))^T(h_1 - h_2) = \mathbb{E}_x[(\nabla_h \ell(x, h_1) - \nabla_h \ell(x, h_2))^T](h_1 - h_2)$$
$$= \mathbb{E}_x[(\nabla_h \ell(x, h_1) - \nabla_h \ell(x, h_2))^T(h_1 - h_2)]$$
$$\geq \mathbb{E}_x[m\|h_1 - h_2\|_\mathbb{F}]$$
$$= m\|h_1 - h_2\|_\mathbb{F},$$

which concludes the proof. $\square$

**Remark 2.** *At this point, it is essential to point out that Assumption 1 is already sufficient for the existence of a global minimizer of $R$ in the supervised learning setting with a finite set of data points. Since we have shown in Lemma 1 that $h \mapsto R(h)$ is m-strong convex as a function on the prediction space. Since the prediction space can be characterized by a finite dimensional set if $R$ only depends on a finite number of values of $h$, we already know that there exists a global minimizer $h^\star \in \mathbb{F}$. However, we do not assume the existence of $\omega^\star \in \mathbb{R}^p$ such that $h(\omega^\star) = h^\star$ (which is not needed for the theoretical findings of this paper). The existence of $\omega^\star$ is generally not guaranteed, since the mapping $\omega \mapsto R(h(\omega))$ does not have to be strongly convex. Later in the numerical experiments as well as the proofs of Corollaries 1 and 2, we assumed the existence of $\omega^\star$, since it holds for the teacher-student setting, and to simplify the calculations.*

As a key feature of analysis in the lazy training regime, the weights $\omega_t$ do not deviate too far from their initial value, which implies that $\lambda Id \preceq Dh(\omega_t)D^T h(\omega_t)$ for some $\lambda > 0$. Many publications have focused on establishing strictly positive lower bounds for this value for deterministic (full-batch) gradient flow (see, for instance, Du et al. (2019b); Chizat et al. (2020)); however, we are not aware of any results that guarantee the positivity of $\lambda = \lambda_{min}(Dh(\omega_t)D^T h(\omega_t))$ for SGD or SGLD. We will solve this problem by introducing a stopping time, which is the smallest time at which this condition no longer holds, and prove in Corollary 3 that this stopping time will be $\infty$ with high probability for $\alpha$ sufficiently large. The minimum eigenvalue of the NTK is strictly positive, at least as long as $||\omega_t - \omega_0||_2 \leq r := \frac{\lambda}{\mathrm{Lip}(Dh)}$. Therefore, we will introduce the random stopping time

$$\tau := \inf\{t \geq 0 : ||\omega_t - \omega_0||_2 > r\},$$

denoting the first time $\lambda Id \preceq Dh(\omega_t)D^T h(\omega_t)$ is degenerate. In addition, the inequality in Assumption 2 also holds at least for $t \leq \tau$. Working with this quantity and proving $\tau = \infty$ with high probability is a major challenge in our analysis. In addition, we define $\min\{T, \tau\} =: \tilde{T}$, where $T > 0$ is some arbitrary but fixed runtime of the algorithm. Let $t < \tilde{T}$ in the following.

**Theorem 1** (Convergence of stochastic gradient Langevin dynamics)**.** *Let $h^\star$ be the global minimizer of $R$. Define the optimality gap as $\bar{R}(h) := R(h) - R(h^\star) > 0$. Then, under Assumptions 1-3, it holds that*

$$\mathbb{E}_\omega[\bar{R}(\alpha h(\omega_t))] \leq \mathbb{E}_\omega[\bar{R}(\alpha h(\omega_0))] \exp(-m\lambda^2 t), \tag{3}$$

*for any $t < \tau$.*

*Proof of Theorem 1.* In the first part, this proof follows Lugosi & Nualart (2024) by using Itô's lemma to find an exponential upper bound of the risk function. In the second part, we use the scaling factor $\alpha$ to control the boundedness of the second-order (Hessian) term by using the interchangeability of the gradient and the expectation of the loss function. This directly connects the boundedness of the Hessian term to the curvature of the loss in parameter space.

As defined, $\omega_t$ follows the SDE in equation 1. In the following, we apply Itô's formula to compute $dg(\omega_t)$, where $g : \mathbb{R}^p \to \mathbb{R}$ is defined by $g(\omega) = \log(\bar{R}(\alpha h(\omega)))$, for all $t$ with $\bar{R}(\alpha h(\omega_t)) > 0$. The gradient and the Hessian of $g$ are as follows:

$$\nabla_\omega g(\omega) = \frac{1}{\bar{R}(\alpha h(\omega))} \alpha D^T h(\omega) \nabla_h R(\alpha h(\omega))$$

$$\nabla_\omega^2 g(\omega) = -\frac{1}{\bar{R}(\alpha h(\omega))^2} \alpha^2 D^T h(\omega) \nabla_h R(\alpha h(\omega)) \nabla_h^T R(\alpha h(\omega)) Dh(\omega) + \frac{1}{\bar{R}(\alpha h(\omega))} \nabla_\omega^2 R(\alpha h(\omega))$$

This implies that $g(\omega_t)$ is the solution of the following SDE

$$dg(\omega_t) = -\frac{\|D^T h(\omega_t) \nabla_h R(\alpha h(\omega_t))\|_2^2}{\bar{R}(\alpha h(\omega_t))} dt \tag{4}$$

$$+ \frac{\eta_\alpha}{2} \mathrm{Tr}\left(-\frac{\|\nabla_h^T R(\alpha h(\omega_t)) Dh(\omega_t) \sigma_\alpha(\omega_t)\|_2^2}{\bar{R}(\alpha h(\omega_t))^2}\right) dt \tag{5}$$

$$+ \frac{\eta_\alpha}{2\alpha^2} \mathrm{Tr}\left(\frac{\sigma_\alpha(\omega_t)^T \nabla_\omega^2 R(\alpha h(\omega_t)) \sigma_\alpha(\omega_t)}{\bar{R}(\alpha h(\omega_s))}\right) dt \tag{6}$$

$$+ \frac{\sqrt{\eta_\alpha} \nabla_h^T R(\alpha h(\omega_t)) Dh(\omega_t) \sigma_\alpha(\omega_t)}{\bar{R}(\alpha h(\omega_t))} dW_t. \tag{7}$$

Similar to Lugosi & Nualart (2024), we observe that the quadratic variation of the process

$$M_t := \sqrt{\eta_\alpha} \int_0^t \frac{\nabla_h^T R(\alpha h(\omega_s)) Dh(\omega_t) \sigma_\alpha(\omega_s)}{\bar{R}(\alpha h(\omega_s))} dW_s,$$

defined in equation 7, is given by

$$\langle M \rangle_t = \eta_\alpha \int_0^t \mathrm{Tr}\left(-\frac{\|\nabla_h^T R(\alpha h(\omega_s)) Dh(\omega_s) \sigma_\alpha(\omega_s)\|_2^2}{\bar{R}(\alpha h(\omega_s))^2}\right) ds,$$

which appears in equation 5. We can use the multi-dimensional Itô formula again on the process $\mathcal{E}_t := e^{M_t - \frac{1}{2}\langle M \rangle_t}$ to prove that it is a nonnegative martingale, which implies that

$$\mathbb{E}_\omega[\mathcal{E}_t] = 1. \tag{8}$$

Returning to equation 4-equation 7, we can rewrite this SDE in the integral form and take exponentials on both sides, which gives

$$\bar{R}(\alpha h(\omega_t)) = \bar{R}(\alpha h(\omega_0)) \exp\left(-\int_0^t \frac{\|D^T h(\omega_s) \nabla_h R(\alpha h(\omega_s))\|_2^2}{\bar{R}(\alpha h(\omega_s))} ds\right)$$

$$\cdot \exp\left(\frac{\eta_\alpha}{2\alpha^2} \int_0^t \mathrm{Tr}\left(\frac{\|\sigma_\alpha(\omega_s)\|_{\nabla_\omega^2 R(\alpha h(\omega_s))}^2}{\bar{R}(\alpha h(\omega_s))}\right) ds + M_t - \frac{1}{2}\langle M \rangle_t\right).$$

Next, we will bound each of the integrals separately. It holds that

$$\int_0^t \frac{\nabla_h^T R(\alpha h(\omega_s)) Dh(\omega_s) D^T h(\omega_s) \nabla_h R(\alpha h(\omega_s))}{\bar{R}(\alpha h(\omega_s))} ds = \int_0^t \frac{\|\nabla_h R(\alpha h(\omega_s))\|_{Dh(\omega_s) D^T h(\omega_s)}^2}{\bar{R}(\alpha h(\omega_s))} ds.$$

Since $t < \tilde{T}$, the eigenvalues of $Dh(\omega_t) D^T h(\omega_t)$ are lower bounded by $\lambda^2$. Therefore, it follows that

$$\int_0^t \frac{\|\nabla R(\alpha h(\omega_s))\|_{Dh(\omega_s) D^T h(\omega_s)}^2}{\bar{R}(\alpha h(\omega_s))} ds \geq \lambda^2 \int_0^t \frac{\|\nabla R(\alpha h(\omega_s))\|_2^2}{\bar{R}(\alpha h(\omega_s))} \geq 2\lambda^2 m \int_0^t ds = 2\lambda^2 mt,$$

using the Polyak-Lojasiewicz inequality, which holds due to the strong convexity of $R$. By Assumption 2, we get

$$\frac{\eta_\alpha}{2\alpha^2} \int_0^t \text{Tr}\left(\frac{\sigma_\alpha(\omega_s)^T \nabla_\omega^2 R(\alpha h(\omega_s)) \sigma_\alpha(\omega_s)}{\bar{R}(\alpha h(\omega_s))}\right) ds \leq \int_0^t \lambda^2 m \, ds$$

Consequently, we obtain

$$\bar{R}(\alpha h(\omega_t)) \leq \bar{R}(\alpha h(\omega_0)) \exp\left(-2\lambda^2 mt + \lambda^2 mt + M_t - \frac{1}{2}\langle M\rangle_t\right)$$

$$= \bar{R}(\alpha h(\omega_0)) \exp\left(-\lambda^2 mt + M_t - \frac{1}{2}\langle M\rangle_t\right)$$

Taking expectations on both sides and using equation 8, the desired result follows. □

**Remark 3.** *It is crucial to point out that Theorem 1 does not provide the result*

$$\mathbb{E}_\omega[\bar{R}(\alpha h(\omega_t))] \xrightarrow{t\to\infty} 0,$$

*but instead*

$$\mathbb{E}_\omega[\bar{R}(\alpha h(\omega_t)) \mathbb{1}_{\{\tau=\infty\}}] \xrightarrow{t\to\infty} 0.$$

*We will return to this problem in Theorem 3 to control the error stemming from $\mathbb{E}_\omega[\bar{R}(\alpha h(\omega_t)) \mathbb{1}_{\{\tau<\infty\}}]$.*

Until now, we have only provided a general result that can be applied to different settings. The following results justify the applicability of Theorem 1 to shallow as well as deep neural networks in supervised learning.

**Corollary 1** (Risk convergence for shallow neural networks). *Consider a shallow neural network of the form*

$$\phi(x;\omega,c) := \frac{1}{\sqrt{m}} \sum_{j=1}^m c_j \sigma(\omega_j^T x_j),$$

*where we only train the weights $\omega$ and randomly choose $c$ at initialization. Here, $\sigma$ is the non-linear activation function $\sigma = \tanh$. This neural network fulfills the assumptions of Theorem 1. This implies that, for appropriate loss functions that fulfill Assumption 1 (such as the mean squared loss function), equation 3 holds for shallow neural networks.*

*Proof of Corollary 1.* The proof is presented in Appendix B. □

In the following, we extend the results for the shallow feedforward neural networks to deep neural networks of depth $H \geq 2$ based on Du et al. (2019b).

**Corollary 2** (Risk convergence for deep neural networks). *Recursively define a H-layer deep neural network*

$$x^{(k)} = \sqrt{\frac{c_\sigma}{m}} \sigma(W^{(k)} x^{(k-1)}), \text{ for } 1 \leq k \leq H$$

$$f(x;\omega) = a^T x^{(H)},$$

*for input data $x^{(0)} \in \mathbb{R}^d$ and $W^{(1)} \in \mathbb{R}^{m\times d}$ as well as $W^{(k)} \in \mathbb{R}^{m\times m}$ for $2 \leq h \leq H$. The parameter $\omega$ contains both the hidden layer weights $W^{(k)}$ as well as the output layer $a$. If $\sigma$ is L-Lipschitz, $\|W^{(k)}(0)\|_2 \leq c_{\omega,0}\sqrt{m}$ and $\|x^{(k)}(0)\|_2 \leq c_{x,0}$, then equation 3 holds for a $\text{Lip}(\nabla\ell)$-smooth and m-strongly convex loss function.*

*Proof.* The proof of Corollary 2 can be found in Appendix C. □

**Remark 4** (First-exit time for deep neural networks)**.** *The theorem cannot be applied directly to deep neural networks, since they do not have a Lipschitz continuous derivative. Therefore we cannot define the first-exit time*

$$\tau := \inf_{0 \leq t}\{t : \|\omega_t - \omega_0\| > \lambda/Lip(Dh)\}.$$

*However, using the Lemmas B1-B4 in Du et al. (2019a), we can define a different first-exit time that guarantees the positive definiteness of the NTK. Instead of using the radius $r := \lambda/Lip(Dh)$, we will use a different radius, for which the above proof can be conducted identically. This circumvents the need for Dh to be Lipschitz-smooth. Further discussion is provided in Appendix C.*

Our immediate next goal is to find an upper bound for the optimality gap (either in predictor or parameter space) that is independent of $\alpha$. This will allow us to increase $\alpha$ without changing the bound itself.

**Corollary 3** (Convergence to the global empirical risk minimizer)**.** *Under Assumptions 1-3 it holds that*

$$\mathbb{E}_{\omega}[\|\alpha h(\omega_t) - h^{\star}\|_{\mathbb{F}}^2] \leq \frac{\text{Lip}(\nabla R)}{m}\mathbb{E}_{\omega}[\|\alpha h(\omega_0) - h^{\star}\|_{\mathbb{F}}^2]\exp(-m\lambda^2 t).$$

*By choosing an initialization for the neural network, such that $h(\omega_0) = 0$, the right-hand side becomes independent of $\alpha$.*

*Proof of Corollary 3.* By the strong $m$-convexity of $R$ follows that

$$\frac{m}{2}\|\alpha h(\omega_t) - h^{\star}\|_{\mathbb{F}}^2 \leq \bar{R}(\alpha h(\omega_t)).$$

We then combine this, with Theorem 1 and the Lipschitz continuity of $R$ to get

$$\frac{m}{2}\mathbb{E}_{\omega}[\|\alpha h(\omega_t) - h^{\star}\|_{\mathbb{F}}] \leq \mathbb{E}_{\omega}[\bar{R}(\alpha h(\omega_0))]\exp(-m\lambda^2 t)$$

$$\leq \frac{\text{Lip}(\nabla R)}{2}\mathbb{E}_{\omega}[\|\alpha h(\omega_0) - h^{\star}\|_{\mathbb{F}}^2]\exp(-m\lambda^2 t),$$

where we used the smoothness inequality $R(\alpha h(\omega_0)) - R(h^{\star}) \leq \frac{1}{2}\text{Lip}(\nabla R)\|\alpha h(\omega_0) - h^{\star}\|_{\mathbb{F}}^2$ by Lemma 3.4 in Bubeck et al. (2015). This leads to the desired equation

$$\mathbb{E}[\|\alpha h(\omega_t) - h^{\star}\|_{\mathbb{F}}^2] \leq \frac{\text{Lip}(\nabla R)}{m}\mathbb{E}_{\omega}[\|\alpha h(\omega_0) - h^{\star}\|_{\mathbb{F}}^2]\exp(-m\lambda^2 t).$$

□

We use the above results to show that by choosing $\alpha$ sufficiently large, we can ensure that $\tau = \infty$ with high probability since $\lambda Id \preceq Dh(\omega_t)D^T h(\omega_t)$ holds. In the following result and throughout the rest of the paper, we denote the conditional probability and expectation given the random initialization $\omega_0$ as $\mathbb{P}_{\omega}$ and $\mathbb{E}_{\omega}$.

**Theorem 2.** *Under Assumptions 1-3, it holds that*

$$\mathbb{P}(\|\omega_t - \omega_0\|_2 > r) \leq \frac{1}{\alpha r}\left(\frac{\|Dh(\omega_0)\|_{\mathbb{F}}\text{Lip}(\nabla R)}{m^{\frac{3}{2}}\lambda^2} + \frac{\text{Lip}(\nabla \ell)}{m\lambda}\right)\sqrt{\text{Lip}(\nabla R)\mathbb{E}_{\omega}[\|h^{\star}\|_2^2]} \lesssim \frac{1}{\alpha}. \tag{9}$$

*Proof of Theorem 2.* To prove the above statement, we will first use equation 1 and rewrite it to get an upper bound on the expected distance of $\omega_t$ from initialization. The resulting terms can then be further bounded by using the regularity of the loss function from Assumption 1 and the results from Theorem 1. We connect the expected distance to the probability of exiting a ball around the initialization using Markov's inequality by $\mathbb{P}(\|\omega_t - \omega_0\| > \epsilon) \leq \frac{\mathbb{E}_{\omega}[\|\omega_t - \omega_0\|]}{\epsilon}$. Using equation 1, we can rewrite the distance of $\omega_t$ to the origin

$$\|\omega_t - \omega_0\|_2 = \left\|\int_0^t -\frac{1}{\alpha}D^T h(\omega_t)\nabla R(\alpha h(\omega_t))dt + \frac{\sqrt{\eta_{\alpha}}}{\alpha}\int_0^t \sigma_{\alpha}(\omega_t)dB_t\right\|_2$$

$$\leq \frac{1}{\alpha}\int_0^t \|D^T h(\omega_t)\|_{\mathbb{F}}\|\nabla R(\alpha h(\omega_t))\|_2 dt + \frac{\sqrt{\eta_{\alpha}}}{\alpha}\|\int_0^t \sigma_{\alpha}(\omega_t)dB_t\|_2$$

Using Cauchy-Schwarz and Theorem 5.21 in Mao (2007) subsequently, we obtain

$$\mathbb{E}_\omega[\|\omega_t - \omega_0\|] \leq \frac{\mathbb{E}_\omega[\int_0^t \|D^T h(\omega_t)\|_\mathbb{F}\|\nabla R(\alpha h(\omega_t))\|_2 dt]}{\alpha} + \frac{\sqrt{\eta_\alpha}\mathbb{E}_\omega[\|\int_0^t \sigma_\alpha(\omega_t)dB_t\|_2]}{\alpha}$$

$$\leq \frac{\mathbb{E}_\omega[\int_0^t \|D^T h(\omega_t)\|_\mathbb{F}\|\nabla R(\alpha h(\omega_t))\|_2 dt]}{\alpha} + \frac{\left[\mathbb{E}_\omega[\int_0^t \text{Tr}(\Sigma_\alpha(\omega_s))ds]\right]^{\frac{1}{2}}}{\alpha}.$$

We will first apply Markov's inequality to obtain

$$\mathbb{P}(\|\omega_t - \omega_0\|_2 \geq r) \leq \frac{\mathbb{E}_\omega[\int_0^t \|D^T h(\omega_t)\|_\mathbb{F}\|\nabla R(\alpha h(\omega_t))\|_2 dt]}{r\alpha} + \frac{\left[\mathbb{E}_\omega[\int_0^t \text{Tr}(\Sigma_\alpha(\omega_s))ds]\right]^{\frac{1}{2}}}{r\alpha}. \quad (10)$$

We will continue by first bounding the first part of the right-hand side in Equation 10.

$$\frac{\mathbb{E}_\omega[\int_0^t \|D^T h(\omega_t)\|_\mathbb{F}\|\nabla R(\alpha h(\omega_t))\|_2 dt]}{r\alpha} \overset{t \leq \tilde{T}}{\leq} \frac{1}{\alpha r}\|Dh(\omega_0)\|_\mathbb{F}\text{Lip}(\nabla R)\mathbb{E}_\omega[\int_0^t \|\alpha h(\omega_t) - h^\star\|_2 dt]$$

$$= \frac{1}{\alpha r}\|Dh(\omega_0)\|_\mathbb{F}\text{Lip}(\nabla R)\int_0^t \sqrt{\mathbb{E}_\omega[\|\alpha h(\omega_t) - h^\star\|_2]^2}dt$$

$$\overset{(\spadesuit)}{\leq} \frac{1}{\alpha r}\|Dh(\omega_0)\|_\mathbb{F}\text{Lip}(\nabla R)\int_0^t \sqrt{\mathbb{E}_\omega[\|\alpha h(\omega_t) - h^\star\|_2^2]}dt$$

$$\overset{(\clubsuit)}{\leq} \frac{1}{\alpha r}\|Dh(\omega_0)\|_\mathbb{F}\text{Lip}(\nabla R)\int_0^t \sqrt{\frac{\text{Lip}(\nabla R)}{m}\mathbb{E}_\omega[\|\alpha h(\omega_0) - h^\star\|_2^2]\exp(-2m\lambda^2 t)}dt$$

$$= \frac{1}{\alpha r}\|Dh(\omega_0)\|_\mathbb{F}\text{Lip}(\nabla R)\sqrt{\frac{\text{Lip}(\nabla R)}{m}\mathbb{E}_\omega[\|\alpha h(\omega_0) - h^\star\|_2^2]}\int_0^t \exp(-m\lambda^2 t)dt$$

$$\leq \frac{1}{\alpha r}\|Dh(\omega_0)\|_\mathbb{F}\text{Lip}(\nabla R)\frac{\sqrt{\text{Lip}(\nabla R)\mathbb{E}_\omega[\|\alpha h(\omega_0) - h^\star\|_2^2]}}{m^{\frac{3}{2}}\lambda^2},$$

where ($\spadesuit$) follows from Jensen's inequality and ($\clubsuit$) follows from Corollary 3. Next, we will bound the second term from the right-hand side of 10. We start by bounding

$$\int_0^t Tr(\Sigma_\alpha(\omega_s))ds = \int_0^t Tr\left(\mathbb{E}_x[\nabla\ell(x, \alpha h(\omega_s)\nabla^T\ell(x, \alpha h(\omega_s))] - R(\alpha h(\omega_s)R^T(\alpha h(\omega_s))\right)ds$$

$$\leq \int_0^t Tr\left(\mathbb{E}_x[\nabla\ell(x, \alpha h(\omega_s)\nabla^T\ell(x, \alpha h(\omega_s))]\right)ds$$

$$= \int_0^t \mathbb{E}_x[\|\nabla\ell(x, \alpha h(\omega_s))\|_2^2]ds$$

$$\overset{(\diamond)}{\leq} \int_0^t \text{Lip}(\nabla\ell)^2\mathbb{E}_x[\|\alpha h(\omega_s) - h^\star\|_\mathbb{F}^2]ds$$

$$\overset{(\clubsuit)}{\leq} \frac{\text{Lip}(\nabla\ell)^2\text{Lip}(\nabla R)}{m}\mathbb{E}_\omega[\|\alpha h(\omega_0) - h^\star\|_\mathbb{F}^2]\int_0^t \exp(-m\lambda^2 s)ds,$$

where we used the smoothness of $\ell$ in $\diamond$ and Corollary 3 in ($\clubsuit$). We can calculate that the value of $\int_0^t \exp(-m\lambda^2 s)ds \leq \frac{1}{m\lambda^2}$, which gives us the following upper bound

$$\frac{\sqrt{\mathbb{E}_\omega\left[\int_0^t Tr(\Sigma_\alpha(\omega_s))ds\right]}}{r\alpha} \leq \frac{\text{Lip}(\nabla\ell)}{r\alpha m\lambda}\sqrt{\text{Lip}(\nabla R)\mathbb{E}_\omega[\|\alpha h(\omega_0) - h^\star\|_\mathbb{F}^2]}.$$

By choosing an initialization such that $h(\omega_0) = 0$, we then get

$$\mathbb{P}(\|\omega_t - \omega_0\|_2 > r) \leq \frac{1}{\alpha r}\left(\frac{\|Dh(\omega_0)\|_\mathbb{F}\text{Lip}(\nabla R)}{m^{\frac{3}{2}}\lambda^2} + \frac{\text{Lip}(\nabla\ell)}{m\lambda}\right)\sqrt{\text{Lip}(\nabla R)\mathbb{E}_\omega[\|h^\star\|_2^2]}.$$

$\square$

In the following corollary, we will show that by choosing $\alpha$ appropriately, $\mathbb{P}(\tau < T)$ is arbitrarily small. This implies that we can effectively use our non-asymptotic bound that we established in Theorem 1 and Corollary 3 to get an upper bound on the training error with arbitrary large probability (depending on $\alpha$).

**Corollary 4.** *Under the same assumptions as Theorem 2, it follows for any $T > 0$*

$$\mathbb{P}(\tau < T) \leq \mathcal{O}(\alpha^{-1}). \tag{11}$$

*Proof of Corollary 4.* In the following let $\tilde{\tau} := \min\{\tau, T\}$. Since the right-hand side of equation 9 does not depend on $t$, we get

$$\begin{aligned}
\mathbb{P}(\tau < T) = \mathbb{P}(\tilde{\tau} < T) \\
\leq \mathbb{P}(\|\omega_{\tilde{\tau}} - \omega_0\| \geq r) \\
\leq \mathcal{O}(\alpha^{-1}),
\end{aligned}$$

where we used Theorem 2 $\square$

The result provided by Corollary 4 is the last ingredient needed to return to the problem introduced in Remark 3 and solve it.

**Theorem 3** ($L^1$-convergence of SGLD)**.** *Under the same assumptions as Theorem 2, it holds that*

$$\mathbb{E}_\omega[\bar{R}(\alpha h(\omega_t))] \leq \exp(-\lambda^2 mt) + \mathcal{O}(\alpha^{-\frac{1}{q}}),$$

*for any $q > 1$ and $t \leq T$ for any $T > 0$.*

*Proof of Theorem 3.* To prove the above result, we will decompose the left-hand side of the result as follows

$$\mathbb{E}_\omega[\bar{R}(\alpha h(\omega_t))] = \mathbb{E}_\omega[\bar{R}(\alpha h(\omega_t)\mathbb{1}_{\{\tau \geq T\}}] + \mathbb{E}_\omega[\bar{R}(\alpha h(\omega_t)\mathbb{1}_{\{\tau < T\}}].$$

Theorem 1 already proved that

$$\mathbb{E}_\omega[\bar{R}(\alpha h(\omega_t)\mathbb{1}_{\{\tau \geq T\}}] \leq \exp(-\lambda^2 mt),$$

which implies that we only need to show

$$\mathbb{E}_\omega[\bar{R}(\alpha h(\omega_t)\mathbb{1}_{\{\tau < T\}}] \leq \mathcal{O}(\alpha^{-\frac{1}{q}}).$$

Using Hölder's inequality with the Hölder conjugate pair $(p, q)$, we get

$$\mathbb{E}_\omega[\bar{R}(\alpha h(\omega_t)\mathbb{1}_{\{\tau < T\}}] \leq \mathbb{E}_\omega[\bar{R}(\alpha h(\omega_t))^p]^{\frac{1}{p}} \mathbb{E}_\omega[\mathbb{1}^q_{\{\tau < T\}}]^{\frac{1}{q}} \tag{12}$$

$$\overset{\spadesuit}{=} \mathbb{E}_\omega[\bar{R}(\alpha h(\omega_t)^p]^{\frac{1}{p}} \mathbb{E}_\omega[\mathbb{1}_{\{\tau < T\}}]^{\frac{1}{q}} \tag{13}$$

$$= \mathbb{E}_\omega[\bar{R}(\alpha h(\omega_t)^p]^{\frac{1}{p}} \mathbb{P}(\tau < T)^{\frac{1}{q}} \tag{14}$$

$$\overset{\clubsuit}{\leq} \mathbb{E}_\omega[\bar{R}(\alpha h(\omega_t)^p]^{\frac{1}{p}} \mathcal{O}(\alpha^{-\frac{1}{q}}), \tag{15}$$

where we used that the image of the characteristic function $\mathbb{1}_{\{\tau < \infty\}}$ is $\{0, 1\}$ in $\spadesuit$, and Corollary 4 in $\clubsuit$. Additionally, it holds that

$$\begin{aligned}
\bar{R}(\alpha h(\omega_t))^p =& \bar{R}(\alpha h(\omega_0))^p \exp\left(-p \int_0^t \frac{\|D^T h(\omega_s)\nabla R(\alpha h(\omega_s))\|_2^2}{\bar{R}(\alpha h(\omega_s))} ds\right) \\
& \exp\left(\frac{p\eta_\alpha}{2\alpha} \int_0^t Tr\left(\frac{\|\sigma_\alpha(\omega_s)\|_{\nabla_\omega^2 R(\alpha h(\omega_s))}}{\bar{R}(\alpha h(\omega_s))}\right) ds\right) e^p \mathcal{E}_t \\
\leq& \bar{R}(\alpha h(\omega_0))^p e^p \mathcal{E}_t,
\end{aligned}$$

where we used an identical calculation to Theorem 1, except for the term $\exp\left(-p \int_0^t \frac{\|D^T h(\omega_s)\nabla R(\alpha h(\omega_s))\|_2^2}{\bar{R}(\alpha h(\omega_s))} ds\right)$, which we upper bounded by 1. Taking the expectation on both sides leads to

$$\mathbb{E}_\omega[\bar{R}(\alpha h(\omega_t))^p] \leq \bar{R}(\alpha h(\omega_0))^p e^p \mathbb{E}_\omega[\mathcal{E}_t] = \bar{R}(\alpha h(\omega_0))^p e^p.$$

Plugging this result into equation 15 concludes the proof. □

Since we have proven in Corollary 4 that, with arbitrarily large probability, $\omega_t$ stays in $B_r(\omega_0)$, we can use this argument to motivate the proximity of the linearized dynamics $\bar{\omega}_t$ to $\omega_t$. As mentioned before, Chizat et al. (2020) uses the scaling factor $\alpha$ to prove the proximity of $h$ to the linearized dynamics defined by $\bar{h}(\omega) := h(\omega_0) + Dh(\omega_0)(\omega - \omega_0)$, which gives rise to the following linear stochastic gradient Langevin dynamics:

$$d\bar{\omega}_t = -\frac{1}{\alpha} D^T h(\omega_0)\nabla R(\alpha\bar{h}(\bar{\omega}_t))dt + \frac{\sqrt{\eta_\alpha}}{\alpha}(\bar{\Sigma}(\bar{\omega}_t))^{\frac{1}{2}}dW_t$$

First, we demonstrate that the above results also apply to the linearized neural network $\bar{\omega}$.

**Corollary 5.** *Assuming that Assumptions 1-3 hold for the non-linearized neural network. In addition, assume that $h(\omega_0) = 0$. Then it follows that*

$$\mathbb{E}_\omega[\|\alpha\bar{h}(\bar{\omega}_t) - h^\star\|_2^2] \leq \frac{\text{Lip}(\nabla R)}{m}\|h^\star\|_2^2 \exp(-m\lambda^2 t), \tag{16}$$

*for all $t \leq \tau$. In addition, it holds that*

$$\mathbb{P}(\|\bar{\omega}_t - \omega_0\|_2 > r) \leq \frac{1}{\alpha r}\left(\frac{\|Dh(\omega_0)\|_\mathbb{F}\text{Lip}(\nabla R)}{m^{\frac{3}{2}}\lambda^2} + \frac{\text{Lip}(\nabla\ell)}{m\lambda}\right)\sqrt{\text{Lip}(\nabla R)\mathbb{E}_\omega[\|h^\star\|_2^2]} = \mathcal{O}\left(\alpha^{-1}\right).$$

*Proof of Corollary 5.* The proof is an application of Corollary 3 and Theorem 2. □

From this, we can then use the triangle inequality to get the following results

**Corollary 6.** *By using the bounds from Corollary 5, we can show that*

$$\mathbb{E}_\omega[\|\alpha\bar{h}(\bar{\omega}_t) - \alpha h(\omega_t)\|_\mathbb{F}] \leq 2\sqrt{\frac{\text{Lip}(\nabla R)}{m}}\|h^\star\|_\mathbb{F}\exp(-m\lambda^2 t).$$

*Proof of Proposition 6.* It holds that

$$\mathbb{E}_\omega[\|\alpha\bar{h}(\bar{\omega}_t) - \alpha h(\omega_t)\|_\mathbb{F}] \leq \mathbb{E}_\omega[\|\alpha\bar{h}(\bar{\omega}_t) - h^\star\|_\mathbb{F}] + \mathbb{E}_\omega[\|\alpha h(\omega_t) - h^\star\|_\mathbb{F}]$$

$$\overset{(\clubsuit)}{\leq} \sqrt{\mathbb{E}_\omega[\|\alpha\bar{h}(\bar{\omega}_t) - h^\star\|_\mathbb{F}^2]} + \sqrt{\mathbb{E}_\omega[\|\alpha h(\omega_t) - h^\star\|_\mathbb{F}^2]}$$

$$\overset{(\spadesuit)}{\leq} 2\sqrt{\frac{\text{Lip}(\nabla R)}{m}}\|h^\star\|_\mathbb{F}\exp(-m\lambda^2 t),$$

where we used Jensen's inequality in ($\clubsuit$), and Corollary 3 and Corollary 5 in ($\spadesuit$).

□

This suggests that, within the lazy training regime, the error between the standard and linearized dynamics decays exponentially over time. For the linearized models, following similar steps, we can obtain equivalent results to Theorem 2 to prove that $\mathbb{P}(\|\omega_t - \bar{\omega}_t\|_2 > \epsilon) \leq \mathcal{O}(\frac{1}{\epsilon\alpha})$ as long as both $\omega_t$ and $\bar{\omega}_t$ are in the lazy training regime.

## 4 Numerical Results

In this section, we apply the results from the last section to a shallow neural network in the teacher-student setting and a deep neural network application. Consider the function defined by

$$\hat{\phi}(x;\omega^\star) := \sum_{j=1}^{m'} c_j^\star \tanh(\omega_j^\star x),$$

where $\omega^\star = [\omega_1^\star \ldots \omega_{m'}^\star] \in \mathbb{R}^{d \times m'}$, $c^\star \in \mathbb{R}^{m'}$, $x \in \mathbb{R}^d$. The training data is generated as

$$y_i = \hat{\phi}(x_i;\omega^\star) + \epsilon_i, \ i = 1, ..., n,$$

where $\omega^\star \in \mathbb{R}^{d \times m'}$ is a predetermined optimum parameter, $\epsilon_i \sim \mathcal{N}(0, Id)$ and $x_i \sim \mathcal{N}(0, Id)$. We consider the empirical mean-squared training error

$$\hat{R}(f) := \frac{1}{n} \sum_{i=1}^{n} (y_i - f(x_i))^2.$$

We consider a single-hidden layer neural network $\phi$ of width $m > m'$ and tanh activations. In order to ensure that $\phi(x;\omega_0) = 0$ for any $x \in \mathbb{R}^d$, we use the centered model $\tilde{\phi}(\cdot;\omega) = \phi(\cdot;\omega) - \phi(\cdot;\omega_0)$, following Chizat et al. (2020). We simulate the stochastic gradient Langevin dynamics using the Euler-Maruyama scheme with a stepsize of $\delta t = 10^{-2}$, a noise factor $\eta_\alpha = 10^{-2}$, and we repeat the experiment for $\alpha \in \{1/8, 8, 32, 256\}$. The input dimension is $d = 16$, and we chose $m = 600 > m' = 1$ as well as a training set of size $n = 800$. We sample the teacher neural network weights from $\omega^\star \sim \text{Unif}([0,1]^{16})$ and choose $c^\star = 1$. The student neural network is initialised by $\omega_j^0 \sim \mathcal{N}(0, Id)$. We only train the hidden layer weights $\omega_j$ and freeze the output layer weights $c_j$ at initialization. The goal of this simulation is to compare the training loss, the minimum eigenvalue of the NTK, and the distance of the weights from initialization ($\sum_{j=1}^{m} \|\omega_j - \omega_j^0\|_2$) for different values of $\alpha$. The proofs, which show that this example fulfills all the needed assumptions, were moved to the appendix. The results are shown in Figure 1, where the training loss rate exhibits an exponential decrease as the value of $\alpha$ increases. Here, we can also see that, even though we have achieved an exponential bound on the training error, this rate is still relatively slow, since the eigenvalues of the NTK are also small. In Figure 2 (left), we can see that the values of $\omega_t$ remain in the vicinity of the initial values for larger $\alpha$. In accordance with this, the minimum eigenvalue of the NTK (right) does decrease faster for smaller values of $\alpha$. For $\alpha = 1/8$, the minimum eigenvalue seems to decrease at a (slow) exponential rate. For the larger values of $\alpha$, this value seems to converge to a value strictly larger than 0, reaffirming our finding in the previous section.

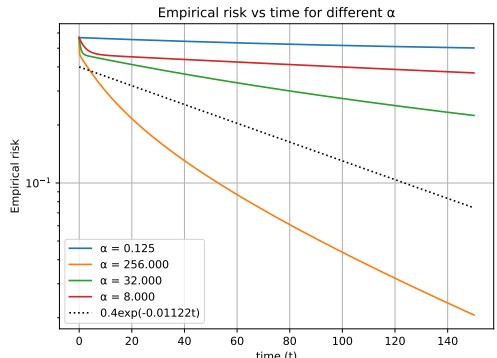

Figure 1: Test-error evolution during training using Euler-Maruyama scheme with stepsize of $\delta t = 0.01$ for $\alpha \in \{1/8, 8, 32, 256\}$. We additionally include curve $f(t)$ with the decay rate $\exp(-\lambda^2 t)$ (dotted line) as a reference, where $\lambda = 0.01122$ is the minimum eigenvalue of the NTK at initialization.

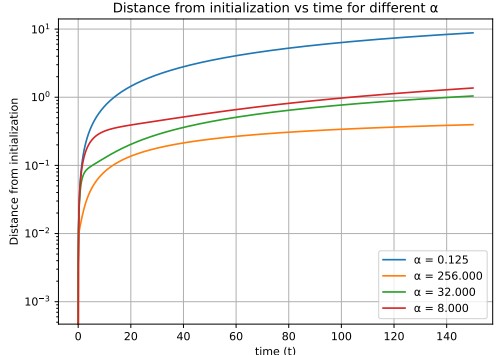 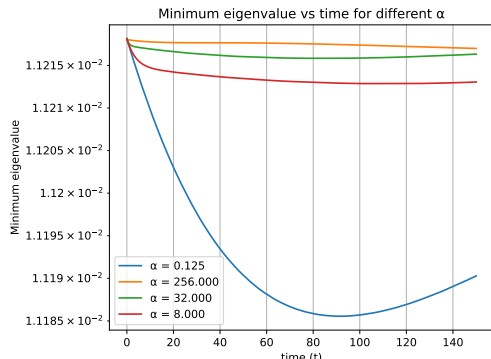

Figure 2: Distance from initialization (left) and value of the minimal eigenvalue of the NTK (right) in time for different values $\alpha \in \{1/8, 8, 32, 256\}$.

## 5   Conclusion

This work analyzed the lazy regime for stochastic gradient Langevin dynamics in the overparameterized setting. First, we proved an exponential convergence rate for the expected optimality gap. Next, we established probabilistic conditions describing the probability of the system leaving this regime. Additionally, we empirically validated these insights in our numerical results. Interesting future research directions include the analysis of SGLD in the underparameterized regime, which was analyzed in Chizat et al. (2020) for the deterministic gradient flow. Another interesting open question is the convergence study of different continuous-time models for SGD in deep learning, such as Li et al. (2017) based on higher-order approximations, and Gurbuzbalaban et al. (2021) that models SGD with heavy-tailed Levy processes.

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

# A    Comparison to Prior Work

Our work is closely related to Lugosi & Nualart (2024) and Chizat et al. (2020). In this section, we explain the main differences and technical challenges in our work compared to these works. The original works in the lazy training regime, such as Chizat et al. (2020); Du et al. (2019b), consider deterministic dynamics, which corresponds to full-batch gradient descent. On the other hand, stochastic gradient descent (SGD) is widely used in practice because of its computational efficiency in large-scale problems. Motivated by this, we analyze continuous-time approximations of SGD by utilizing Itô stochastic differential equations (SDEs), which makes the analysis substantially different and more challenging compared to the ODEs studied in Chizat et al. (2020). To be more concrete, the dynamics presented in this previous work can be rewritten as

$$d\omega_t = -\frac{1}{\alpha}D^T h(\omega_t)\nabla_h R(\alpha h(\omega_t))dt.$$

The dynamics that are treated in this publication

$$d\omega_t = -\frac{1}{\alpha}D^T h(\omega_t)\nabla_h R(\alpha h(\omega_t))dt + \frac{\sqrt{\eta_\alpha}}{\alpha}(\Sigma_\alpha(\omega_t))^{\frac{1}{2}}dW_t,$$

include the additional stochastic term that models subsampling in stochastic learning algorithms such as stochastic gradient descent. The analysis in Chizat et al. (2020) is not directly applicable to the stochastic setting due to the diffusion term that accounts for stochasticity, which requires different and more sophisticated analysis tools to control the diffusion term. On the other hand, Lugosi & Nualart (2024) considers stochastic dynamics similar to our setting. The key difference in our work is that we analyze the training dynamics of deep neural networks in the lazy training regime, characterized by an output scaling factor $\alpha > 0$ that controls the parameter movement. This scaling factor and the analysis of the stopping time $\tau$ (which is a specific case of $\tau_r$ in Lugosi & Nualart (2024)) describing the exit time of the lazy training regime, gives us a strictly positive lower bound on $A_{\min}(r, \omega_0)$ which loosens the restriction on $\theta(r, \omega_0, \eta)$ in Lugosi & Nualart (2024). Additionally, by formulating the problem immediately as a supervised learning problem, by considering $\omega \mapsto R(h(\omega))$ instead of $\omega \mapsto f(\omega)$ combined with the lazy training formulation, we make directly verifiable assumptions on the loss function and neural network model in a supervised learning setting, while the assumptions in Lugosi & Nualart (2024) depend on more complicated problem parameters. Overall, this slight change to the setting yields more relaxed, easier-to-verify assumptions that achieve similar results.

A new version of Lugosi & Nualart (2024) has since been published, which includes similar results for non-linear deep neural networks as well. There are two additional technical differences that we want to point out. First, for most of our results, we do not have to assume the existence of a global minimizer in the parameter space $\omega^\star$ (see Remark 2). Secondly, the comparison between the stochastic Langevin dynamics and the linearized dynamics in Corollary 5 and 6 as well as separated analysis in the function space (Theorem 2) has not been conducted in the stochastic setting.

| | Chizat et al. (2020) | Lugosi & Nualart (2024) | Lugosi & Nualart (2025) | Our work |
|---|---|---|---|---|
| stochastic dynamics | | $\times$ | $\times$ | $\times$ |
| lazy training | $\times$ | | | $\times$ |
| non-linear DNN | $\times$ | | $\times$ | $\times$ |
| linear comparisson | $\times$ | | | $\times$ |

Table 1: Illustration of the key differences between our work and previous works.

# B    Applicability Proofs for Shallow Neural Networks

In this section, we will provide the proofs needed to show that the example from Section 4 fulfills Assumption 1 and 2. Assumption 3 follows immediately, since we are in the supervised learning setting, where the sample measure is discrete.

**Proposition 1** (Smoothness and strong convexity of $\ell$)**.** *The loss-function*

$$\ell(x, h) = (y - h(x))^2$$

*is strongly convex and its gradient (wrt. h) is Lipschitz continuous with respect to h.*

*Proof.* The gradient of the loss function is given by

$$\nabla_h \ell(x, h) = \nabla_h \|h(x) - y\|_2^2 = 2(h(x) - y).$$

We first prove the strong convexity

$$
\begin{aligned}
(\nabla_h \ell(x, h_1) - \nabla_h^T \ell(x, h_2))(h_1(x) - h_2(x)) &= 2(h_1(x) - h_2(x))^T (h_1(x) - h_2(x)) \\
&= 2\|h_1(x) - h_2(x)\|_2^2.
\end{aligned}
$$

The Lipschitz continuity of the gradient follows immediately from the calculation of the gradient

$$\|\nabla_h \ell(x, h_1) - \nabla_h \ell(x, h_2)\|_2 = 2\|h_1(x) - h_2(x)\|_2$$

□

Next, we will prove the Lipschitz continuity of $D_\omega \phi$.

**Proposition 2** (Lipschitz continuity of $D_\omega \phi$)**.** *The derivative $D_\omega \phi$ of*

$$\phi(x; \omega, c) := \frac{1}{\sqrt{m}} \sum_{j=1}^{m} c_j \sigma(\omega^T x),$$

*where $\sigma = \tanh$ is Lipschitz continuous with respect to $\omega$.*

*Proof.* The derivative of $\phi$ with respect to $\omega_i$ is given by

$$D_{\omega_i} \phi(x; \omega, c) := \frac{c_i}{\sqrt{m}} \sigma'(\omega_i^T x) x^T.$$

We know that the derivative of tanh is Lipschitz continuous. It follows that

$$
\begin{aligned}
\|D_{\omega_i} \phi(x; \omega, c) - D_{\omega_i} \phi(y; \tilde{\omega}, c)\| &\leq \frac{1}{\sqrt{m}} |c_i| \|\sigma'(\omega_i^T x) - \sigma'(\tilde{\omega}_i^T x)\| \|x\| \\
&\leq \frac{\mathrm{Lip}(\sigma')}{\sqrt{m}} |c_i| \|x\| \|\omega_i^T x - \tilde{\omega}_i^T x\| \\
&= \frac{\mathrm{Lip}(\sigma')}{\sqrt{m}} |c_i| \|x\|^2 \|\omega_i - \tilde{\omega}_i\|,
\end{aligned}
$$

which proves the Lipschitz continuity.

□

The only requirement, that still needs to be proven is Assumption 2. In order to do this, we will split the proof in two parts. First we will prove the following helpful lemma.

**Lemma 2.** *[Curvature of parameter-loss-mapping] There exist parameters $\alpha, \eta_\alpha > 0$ such that $\nabla_\omega^2 (R(\alpha h(\omega))) \preceq \frac{\alpha^2}{\eta_\alpha} Id.$*

*Proof.* To prove this property, we need to calculate the Hessian of the mapping $\omega \mapsto R(\alpha h(\omega))$ and then determine the maximum eigenvalue of the Hessian. To do this, we will first decompose the Hessian into different parts, and then write the eigenvalues of the components using block-submatrices. The gradient of the above function is given by

$$\nabla_\omega R(\alpha \phi(x; \omega, c)) = \frac{\alpha}{n} \sum_{i=1}^{n} r_i(\omega) J_i(\omega),$$

where

$$r_i(\omega) := \alpha\phi(x_i; \omega, c) - y_i$$
$$J_i(\omega) = \nabla_\omega\phi(x_i; \omega, c).$$

Using this notation, we can write the Hessian as

$$\nabla_\omega^2 R(\alpha\phi(x; \omega, c)) = \frac{\alpha^2}{n}\sum_{i=1}^n J_i(\omega)J_i(\omega)^T + \frac{\alpha}{n}\sum_{i=1}^n r_i(\omega)\nabla_\omega^2\phi(x; \omega, c),$$

which implies that

$$\lambda_{max}\left(\nabla_\omega^2 R(\alpha\phi(x; \omega, c))\right) \leq \frac{\alpha^2}{n}\sum_{i=1}^n \|J_i(\omega)\|^2 + \frac{\alpha}{n}\sum_{i=1}^n |r_i(\omega)|\|\nabla_\omega^2\phi(x; \omega, c)\|.$$

In other words, we only need to bound $\|J_i(\omega)\|, |r_i(\omega)|$ and $\|\nabla_\omega^2\phi(x_i; \omega, c)\|$ in order to show the proposition.

- We start off by bounding $\|J_i(\omega)\|^2$:

$$\begin{aligned}
\|J_i(\omega)\|^2 &= \sum_{j=1}^m \|\frac{c_j}{\sqrt{m}}\sigma'(\omega_j^T x_i)x_i\|^2 \\
&= \frac{\|x_i\|^2}{m}\sum_{j=1}^m c_j^2\sigma'(\omega_j^T x_i)^2 \\
&\leq \frac{\|x_i\|^2}{m}\|c\|_2^2
\end{aligned}$$

where we used $\forall x \in \mathbb{R} : |\sigma'(x)| \leq 1$.

- The bound of $r_i$ follows immediately from

$$|r_i(\omega)| = |\alpha\phi(x_i; \omega, c) - y_i| \leq \alpha|\frac{1}{\sqrt{m}}\sum_{j=1}^m c_j\sigma(\omega_j^T x)| + |y_i| \leq \frac{\alpha}{\sqrt{m}}\|c\|_1 + |y_i|.$$

- Lastly, we still need to bound $\|\nabla_\omega^2\phi(x_i; \omega, c)\|$. We will do this by considering the blocks of the Hessian $\nabla_{\omega_j,\omega_k}^2\phi(x_i; \omega, c)$. First, we observe for $j \neq k$

$$\nabla_{\omega_j,\omega_k}^2\phi(x_i; \omega, c) = D_{\omega_k}\left(\frac{1}{\sqrt{m}}c_j\sigma(\omega_j^T x)x\right) \overset{j \neq k}{=} 0,$$

which means that this Hessian is a block-diagonal matrix. In addition, it holds for $j = k$

$$\nabla_{\omega_j,\omega_j}^2(\phi(x_i; \omega, c)) = \frac{1}{\sqrt{m}}c_j\sigma''(\omega_j^T x_i)x_i x_i^T.$$

Taking the norm of this equation, we receive

$$\lambda_{max}\left(\nabla_{\omega_j,\omega_j}^2(\phi(x_i; \omega, c))\right) \leq \frac{4}{3\sqrt{3m}}c_j\|x_i\|,$$

where we used $\forall x \in \mathbb{R} : \|\sigma''(x)\| \leq \frac{4}{3\sqrt{3}}$.

Putting all of the above together, we get the upper bound

$$\lambda_{max}(\nabla_\omega^2 R(\alpha\phi(x; \omega, c))) \leq \frac{\alpha^2\|c\|_2^2\|x\|_2^2}{mn} + \frac{4\alpha\|c\|_\infty}{3n\sqrt{3m}}\left(\frac{\alpha\|c\|_1}{\sqrt{m}} + \|y\|_\infty\right)\|x\|_2^2,$$

which is in $\mathcal{O}(\alpha^2)$ for a given dataset and predictor class. This implies that for $\eta_\alpha$ sufficiently small, the above proposition holds. $\qquad\square$

The following proposition will rely on the above Lemma 2 to show that Assumption 2 holds for shallow neural networks trained on a finite training set with the mean squared error as the loss function.

**Proposition 3.** *In the above shallow neural network setting it holds that*

$$Tr\left(\frac{\sigma_\alpha(\omega_s)^T \nabla_\omega^2 R(\alpha h(\omega_s))\sigma_\alpha(\omega_s)}{\bar{R}(\alpha h(\omega_s))}\right) \leq \frac{2\alpha^2\lambda^2 m}{\eta_\alpha}.$$

*This result effectively justifying the application of Theorem 1 to shallow neural networks.*

*Proof.* In the following, we will assume that the global minimizer does have zero residual risk, i.e. $R(h^\star) = 0$. This implies that $\bar{R}(\alpha h(\omega_s)) = R(\alpha h(\omega_s))$. By using the basic property of the Trace operator, we get

$$Tr\left(\frac{\sigma_\alpha(\omega_s)^T \nabla_\omega^2 R(\alpha h(\omega_s))\sigma_\alpha(\omega_s)}{R(\alpha h(\omega_s))}\right) = Tr\left(\frac{\Sigma_\alpha(\omega_s)\nabla_\omega^2 R(\alpha h(\omega_s))}{R(\alpha h(\omega_s))}\right)$$

$$\leq Tr\left(\frac{\Sigma_\alpha(\omega_s)}{R(\alpha h(\omega_s))}\right) Tr(\nabla_\omega^2 R(\alpha h(\omega_s))).$$

From Lemma 2 we know how to bound $Tr(\nabla_\omega^2 R(\alpha h(\omega_s)))$. For the other trace operator, we can proceed as follows:

$$Tr\left(\frac{\Sigma_\alpha(\omega_s)}{R(\alpha h(\omega_s))}\right) = Tr\left(\frac{\mathbb{E}_\omega[\nabla_h\ell(x,\alpha h(\omega_s))\nabla_h^T\ell(x,\alpha h(\omega_s))] - \nabla_h R(\alpha h(\omega_s))\nabla_h^T R(\alpha h(\omega_s))}{R(\alpha h(\omega_s))}\right)$$

$$\leq Tr\left(\frac{\mathbb{E}_\omega[\nabla_h\ell(x,\alpha h(\omega_s))\nabla_h^T\ell(x,\alpha h(\omega_s))]}{R(\alpha h(\omega_s))}\right)$$

$$= \mathbb{E}_\omega\left[\frac{Tr(\nabla_h\ell(x,\alpha h(\omega_s))\nabla_h^T\ell(x,\alpha h(\omega_s)))}{R(\alpha h(\omega_s))}\right]$$

$$= \mathbb{E}_\omega\left[\frac{\|\nabla_h\ell(x,\alpha h(\omega_s))\|_2^2}{R(\alpha h(\omega_s))}\right].$$

Next we use, that $\ell(x,\alpha h(\omega_s)) = (y - \alpha h(\omega_s))^2$, to get

$$\mathbb{E}_\omega\left[\frac{\|\nabla_h\ell(x,\alpha h(\omega_s))\|_2^2}{R(\alpha h(\omega_s))}\right] = 4\mathbb{E}_\omega\left[\frac{\|(y - \alpha h(\omega_s))\|_2^2}{R(\alpha h(\omega_s))}\right] = 4\frac{R(\alpha h(\omega_s))}{R(\alpha h(\omega_s))} = 4.$$

In total, we get

$$Tr\left(\frac{\sigma_\alpha(\omega_s)^T \nabla_\omega^2 R(\alpha h(\omega_s))\sigma_\alpha(\omega_s)}{R(\alpha h(\omega_s))}\right) \leq \mathcal{O}(\alpha^2),$$

which means that Proposition 3 holds for appropriate $\eta_\alpha$. □

## C  Applicability Proofs for Deep Neural Networks

The goal of this section is to provide evidence that the results from this publication can also be used for deep neural networks under reasonable assumptions. As already explained in Remark 4, we can circumvent the need of Lipschitz continuity of $Dh$ by using a different stopping time that allows an identical proof to Theorem 1 by guaranteeing the positive definiteness of $\lambda_{min}(Dh(\omega_t)D^T h(\omega_t))$. Since we heavily rely on the results from Du et al. (2019a), we will only consider the same class of deep neural networks as those presented in the original publication. Let $x^{(0)} \in \mathbb{R}^d$ be the input, $W^{(1)} \in \mathbb{R}^{m\times d}$ be the first weight matrix, $W^{(k)} \in \mathbb{R}^{m\times m}$ the weight at the k-th layer, for layers $2 \leq k \leq H$. $a \in \mathbb{R}^m$ is the output layer weights and $\sigma(\cdot)$ is an activation function. Additionally, define the normalizing scaling factor $c_\sigma := (\mathbb{E}_{x\sim N(0,1)}[\sigma(x)^2])^{-1}$. Then we recursively define

$$x^{(k)} = \sqrt{\frac{c_\sigma}{m}}\sigma(W^{(k)}x^{(k-1)}), \text{ for } 1 \leq k \leq H \tag{17}$$

$$f(x,\omega) = a^T x^{(H)}, \tag{18}$$

and $x^{(0)} = x$ is the input of the neural network. We will present the results from Du et al. (2019a) in the following propositions. Although the original publication uses discrete time-steps, contrary to the continuous dynamics presented in this text, the proofs of Lemmas 1-4 in Appendix B never explicitly use the discrete time steps. Therefore, the proofs hold analogously in the continuous setting. We will only present Lemmas 3 and 4 in the following. The geometric series function $g_\alpha(n) = \sum_{i=0}^{n-1} \alpha^i$ will be used extensively.

**Proposition 4** (Bound on output difference; Lemma B.3 in Du et al. (2019a)). *Suppose for every $k \in \{1, ..., H\}$, $\|W^{(k)}(0)\|_2 \leq c_{w,0}\sqrt{m}$, $\|x^{(k)}(0)\|_2 \leq c_{x,0}$ and $\|W^{(k)}(t) - W^{(h)}(0)\|_{\mathbb{F}} \leq \sqrt{m}R$ for constants $c_{w,0}, c_{x,0} > 0$ and $R \leq c_{w,0}$. If $\sigma(\cdot)$ is L-Lipschitz, we have*

$$\left\| x^{(k)}(t) - x^{(k)}(0) \right\|_2 \leq \sqrt{c_\sigma} L c_{x,0} g_{c_x}(k) R,$$

*where $c_x = 2\sqrt{c_\sigma} L c_{w,0}$.*

*Proof.* See Du et al. (2019a) Appendix B. $\square$

The condition that we still need to prove is $\lambda_{min}(D^T h(\omega_t) Dh(\omega_t)) \geq \lambda$. To do this, we will look at the Gram matrix $G(k) \in \mathbb{R}^{n \times n}$, defined by

$$G_{i,j}(s) = \left\langle \frac{\partial f(\omega_s, x_i)}{\partial \omega_s}, \frac{\partial f(\omega_s, x_j)}{\partial \omega_s} \right\rangle$$

$$= \sum_{k=1}^{H} \left\langle \frac{\partial f(\omega_s, x_i)}{\partial W_s^{(k)}}, \frac{\partial f(\omega_s, x_j)}{\partial W_s^{(k)}} \right\rangle + \left\langle \frac{\partial f(\omega_s, x_i)}{\partial a_s}, \frac{\partial f(\omega_s, x_j)}{\partial a_s} \right\rangle$$

$$=: \sum_{k=1}^{H+1} G_{i,j}^{(k)}(s).$$

Each $G^{(k)}(s)$ is a positive semi-definite matrix. Therefore, we can bound the minimal Eigenvalue by only considering $G^{(H)}(s)$.

**Proposition 5** (Lemma B.4 in Du et al. (2019a)). *Suppose $\sigma(\cdot)$ is L-Lipschitz and $\beta$-smooth. Suppose for $k \in \{1, ..., H\}$, $\|W^{(k)}(0)\|_2 \leq c_{w,0}\sqrt{m}$, $\|a(0)\|_2 \leq a_{2,0}\sqrt{m}$, $\|a(0)\|_4 \leq a_{4,0}m^{1/4}$, $\frac{1}{c_{x,0}} \leq \|x^{(k)}(0)\|_2 \leq c_{x,0}$, if $\|W^{(k)}(s) - W^{(k)}(0)\|_{\mathbb{F}}, \|a(s) - a(0)\|_2 \leq \sqrt{m}R$ where $R \leq cg_{c_x}(H)^{-1}\lambda n^{-1}$ and $R \leq cg_{cx}(H)^{-1}$ for some small constant $c$ and $c_x = 2\sqrt{c_\sigma} L c_{w,0}$, we have*

$$\|G^{(H)}(s) - G^{(H)}(0)\|_2 \leq \frac{\lambda}{4}.$$

*Proof.* See Du et al. (2019a) Appendix B. $\square$

Under the assumption that $\lambda_{min}(D^T h(\omega_0) Dh(\omega_0)) \geq \lambda$, this gives us the needed result to define a new stopping time on the neural network weights. Instead of considering

$$\tau := \inf\{t \geq 0 : \|\omega_t - \omega_0\| > \lambda/\text{Lip}(Dh)\},$$

we need to consider the stopping time $\tau := \inf_{k \in \{0,...,H\}} \tau^{(k)}$ with

$$\tau^{(k)} = \begin{cases} \inf\{t \geq 0 : \|W^{(k)}(t) - W^{(k)}(0)\|_{\mathbb{F}} > \sqrt{m}R\}, & \text{if } k > 0 \\ \inf\{t \geq 0 : \|a(t) - a(0)\|_2 > \sqrt{m}R\}, & \text{if } k = 0. \end{cases}$$

By using this stopping time, we can derive a similar result to Theorem 1 for deep neural networks. The following lemma will show that Assumption 2 is realistic for deep neural networks, by proving its validity for supervised learning with the mean squared loss function.

**Lemma 3.** *We consider the deep neural network defined in Equations 17 and 18 with $\sigma = \tanh$ as its activation function. Additionally, we consider the loss function*

$$\ell(x, h) = (y - h(x))^2.$$

*Then Assumption 2 holds, i.e.,*

$$Tr\left(\frac{\sigma_\alpha(\omega)^T \nabla_\omega^2 R(\alpha h(\omega))\sigma_\alpha(\omega)}{\bar{R}(\alpha h(\omega))}\right) \leq \frac{2\alpha^2 \lambda^2 m}{\eta_\alpha},$$

*for all $\omega \in B_r(\omega_0)$.*

*Proof.* In Appendix B we have already shown that this loss function and the expected loss $R(h)$ fulfill Assumption 1 (Proposition 1). Assumption 3 follows again, since we consider a discrete probability measure on the finite set of training data. Similar to Proposition 3 to get

$$Tr\left(\frac{\sigma_\alpha(\omega)^T \nabla_\omega^2 R(\alpha h(\omega))\sigma_\alpha(\omega)}{\bar{R}(\alpha h(\omega))}\right) \leq Tr\left(\frac{\Sigma_\alpha(\omega)}{\bar{R}(\alpha h(\omega))}\right) Tr\left(\nabla_\omega^2 R(\alpha h(\omega))\right).$$

We can use the same argument to bound the first trace operator as we did in Proposition 3. Additionally, we will bound the trace of the hessian as follows:

$$\nabla_\omega^2 R(\alpha h(\omega)) = \nabla_\omega^2 \frac{1}{n} \sum_{i=1}^n \ell(x_i, \alpha h(\omega))$$

$$= \frac{1}{n} \sum_{i=1}^n \nabla_\omega^2 \ell(x_i, \alpha h(\omega)),$$

where we used that we only consider the expectation over a finite set of data points $\{(x_i, y_i)\}_{i=1}^n$. It holds that

$$\nabla_\omega^2 \ell(x, \alpha h(\omega)) = D(\nabla_\omega(y - \alpha h(\omega)(x))^2)$$

$$= D(2(y - \alpha h(\omega)(x))(-\alpha \nabla_\omega h(\omega)(x)))$$

$$= 2(\alpha^2 \nabla_\omega h(\omega)(x) \nabla^T h(\omega)(x) + \alpha(\alpha h(\omega)(x) - y)\nabla_\omega^2 h(\omega)(x)).$$

Taking the trace operator, we get

$$Tr(\nabla_\omega^2 R(\alpha h(\omega))) \leq \frac{2}{n} \sum_{i=1}^n (\alpha^2 \|\nabla_\omega h(\omega)\|_2^2 + \alpha(\alpha h(\omega)(x) - y)Tr(\nabla_\omega^2 h(\omega))).$$

Next, we can easily show that this trace is in $\mathcal{O}(\alpha^2)$, since

$$|\alpha h(\omega)(x) - y| \leq \alpha|h(\omega)(x)| + |y| \leq \alpha \max_{x=x_1,\dots,x_n} |h(\omega)(x)| + \max_{y=y_1,\dots,y_n} |y|,$$

and $\omega \mapsto h(\omega)(x)$ for a given $x$, is a $C^2$ mapping by construction, it takes a maximum value on $\bar{B}_r(\omega_0)$. By the same argument, $\nabla_\omega h(\omega)(x)$ and $\nabla_\omega^2 h(\omega)(x)$ are bounded on $\bar{B}_r(\omega_0)$ for a given $x$. This implies that

$$Tr(\nabla_\omega^2 R(\alpha h(\omega))) \leq \frac{2}{n} \sum_{i=1}^n (\mathcal{O}(\alpha^2) + \mathcal{O}(\alpha^2)) = \mathcal{O}(\alpha^2),$$

which proofs Lemma 3 for sufficiently large $\eta_\alpha$. □

**Remark 5.** *In the proof of Lemma 3, we used crude upper bounds to show the validity of Assumption 2. For a given architecture, loss function and data distribution one can easily compute the exact hessian and find explicit upper bounds that result in direct bounds for minimal choices of $\frac{\alpha^2}{\eta_\alpha}$.*

