# OpenReview forum: "Convergence of Stochastic Gradient Langevin Dynamics in the Lazy Training Regime"
_TMLR — Accepted by TMLR_

### Review · Reviewer_1uFt · 2025-11-02

**Summary Of Contributions:**

This paper provides a rigorous, non-asymptotic convergence analysis of stochastic gradient Langevin dynamics (SGLD) viewed as a continuous-time Itô stochastic differential equation approximation of stochastic gradient descent. Focusing on the lazy training regime, the authors prove that under suitable regularity assumptions on the Hessian of the loss function, SGLD with multiplicative, state-dependent noise:

- Maintains a non-degenerate neural tangent kernel throughout training with high probability, ensuring stable dynamics;

- Achieves exponential convergence in expectation to the empirical risk minimizer, with explicit finite-time and finite-width bounds on the optimality gap.

- The theoretical results are supported by numerical experiments in a regression setting, illustrating the practical relevance of the analysis.

**Audience:**

Yes

**Audience Explanation:**

The TMLR community would be interested in this work because it provides rigorous, non-asymptotic convergence guarantees for SGLD in a realistic lazy training regime, incorporating multiplicative, state-dependent noise. By connecting the dynamics to the neural tangent kernel and establishing finite-time, finite-width bounds, the paper advances theoretical understanding of optimization in deep learning.

Contrary to previous works, that focused on establishing strictly positive lower bounds on the eigenvalues of the Neural Tangent Kernel for full-batch gradient flow, the paper extends those results to SGD.

**Broader Impact Concerns:**

I don't see any concerns on the ethical implications of the work.

**Claims And Evidence:**

Yes

**Claims Explanation:**

I thank the authors for their work. The paper is overall well written. The results are new and interesting for the community. The theoretical results are supported by numerical simulations.

In my opinion, the work merits publication. However, I have identified some possible errors in the proofs, so I recommend that the authors address these issues before the paper is accepted. Please refer to my comment in the "Requested Changes" section.

**Requested Changes:**

Here are my major concerns:

- I might have found a wrong statement in the proof of Thoerem 2. The authors claim on the top of page 9 that $\int_0^t \mathrm{Tr}(\Sigma_{\alpha}(\omega_s))ds \leq 0$, but this term appear with a square root on the line above. This is problematic in case the integral is negative. Could the authors correct/clarify this part of the proof ?

- Could the authors clarify why it is licit to write $\mathbb{P}(\tau <\infty) \leq \mathbb{P}(\|\omega_{\tau}-\omega_0\|>r)$ in the proof of Corollary 4 ? More precisely, why the object $\omega_{\tau}$ is well-defined when $\tau=\infty$ ?


Here are some minor comments:

- It would be helpful to ensure consistent notation. For example, the expectation operator is sometimes written with a subscript $x$ and sometimes not (see Assumption 3 and the proof of Lemma 1). Similarly, it would help to specify the variable with respect to which the gradient is taken in the nabla notation. This is done in Assumption 2, but not in Assumptions 1 and 3.

- In Section 2.1, should $R$ satisfy some properties to ensure that $\omega^*$ is well defined (i.e., that a global minimizer exists)?

- page 5: "stray" should be stay.

- I don't understand the use of Dh: it seems the authors use this notation as if it referred to the gradient associated (while based page 2, it refers to the differential). Why not maintain consistency and use the ∇ (nabla) notation instead?

- There are some inconsistencies in the notation: sometimes D^{\top}h is used (e.G. in section 2.2), sometimes Dh^{\top} is used (e.g. in assumption 2)

- In Corollary 1, the notation \sigma is not defined

- In Eq.9, there is typo regarding the style of "F" in the subscript of the norm. Same in the statement and the proof of Theorem 2.

- I think there are some inconsistencies in the paper regarding the use of either $\| \cdot \|_2$ or $\| \cdot \|_{\mathbb {F}}$. For example, in $h(\omega)$ should be in $\mathbb {F}$, but the norm $\| \cdot \|_2$  is often used for those objects such as in the statement and proof of Corollary 3.

- There is a typo at the end of the proof of Theorem 3 with "proofn".

- In Section 4, $c^{\star}$ should be in $\mathbb{R}^{m'}$ and not $\mathbb R$. Also, is a transpose missing in $\omega_j^* x$ ?

---

> ### Author Response · Authors · 2025-12-03
>
> We thank the reviewer for taking the time to carefully read our paper and for providing very valuable and constructive feedback. Below, we address the questions and concerns raised by the reviewer. In the revised manuscript, all changes are highlighted in blue.
>
> **Technical error in Theorem 2.** Thank you for identifying this issue. In the revised manuscript, we corrected the argument. The error stemmed from the incorrect use of the Lipschitz inequality in Theorem 1. We fixed it using a slightly strengthened form of Assumption 2 (page 4), which still automatically holds for both shallow and deep neural networks, as shown in Proposition 3 and Lemma 3. We apologize for the oversight and thank the reviewer for pointing out the error.
>
> **Definition of $\omega_\infty$.** The previous use of $\omega_\infty$ was mathematically imprecise, as convergence of $(\omega_t)$ was not established. We resolved this by reformulating the statement as a non-asymptotic result valid for any terminal time $T>0$ (page 10). Stopped processes with time-horizon $T$, similar to our setting, is standard in SDEs.
>
> **Notation of $Dh$.** To address the reviewers' comment, we have updated the notation section to clarify the meaning of $Dh$. In the supervised learning setting with finite data, it corresponds to the Jacobian; if $h : \mathbb{R}^d \to \mathbb{R}$, it is the transpose of the gradient.
>
> **Existence of $\omega^\ast$.** For our main results, the existence of a global minimizer in parameter space is not required. We only rely on the existence of a global minimizer in the (typically larger) Hilbert space $\mathbb{F}$, which is guaranteed in the finite-data supervised learning setting due to $m$-strong convexity. The application results for shallow and deep neural networks use the existence of $\omega^\star$, which is justified in the teacher-student setting. We clarify this in Remark~2. Additionally, we reformulated the original problem setting to not contain $\omega^\star$.
>
> **Other comments.** We have addressed all comments regarding notation and typographical issues. We thank the reviewer.

---

> > ### Comment · Reviewer_1uFt · 2026-01-10
> >
> > I thank the authors for carefully reading my review. All the points raised have been adequately addressed, and I appreciate in particular the addition of Remark 2 in the revised manuscript, which clarifies one of my questions.
> >
> > After reading the revised version, I only noticed a few remaining typographical issues:
> >
> > - The sentence ``which appears in equation 6'' should instead refer to equation 5.
> > - On page 5, there is a typographical error where ``leqr'' appears in the main text.
> > - At page 9, there is a typo in " second therm".
> > - At the top of page 16, in the definition of $r_i(\omega)$, should $y_i$ be used instead of $y_n$?
> > - On page 16, when bounding $\|\|J_i(\omega)\|\|^2$, the variable should be $x_i$ rather than $x$. Moreover, in the second equality, $w_j$ should be used instead of $w_i$.
> >
> > In my opinion, the paper is now ready for publication.

---

> > > ### Author Response · Authors · 2026-01-29
> > >
> > > We thank the reviewer for carefully rereading the manuscript and for pointing out the remaining typographical issues. We have corrected all of them in the revised version.

---

### Review · Reviewer_Gbbb · 2025-11-13

**Summary Of Contributions:**

The paper considers the question of performing a non-asymptotic convergence analysis of stochastic gradient Langevin dynamics in the lazy training regime. Three main results are given: 1) an inequality that establishes an exponential decay of the expected gap between the loss of the solution obtained by SGLD and the optimal loss, 2) a concentration inequality which shows that, with high probability, the model will not deviate significantly from the initial state, thus staying in the lazy training regime (which also translates to an upper bound on the expected optimality gap), and 3) applications to obtain upper bounds on the training loss of neural networks with finite depth and width.

I believe this is generally a decent paper. The results are conceptually interesting, with the assumptions involved in them not appearing to be overly restrictive in the context of the overall literature. I guess one weak point could be the fact that there appears to be strong reliance to both Chizat et. al. (2020), and Lugosi and Nualart (2024). In that sense, the paper may not be the strongest in terms of technical novelty. For that reason, I'm interested in reading the comments by the authors where they discuss this aspect of the work.

Lenaic Chizat, Edouard Oyallon, and Francis Bach. On lazy training in differentiable programming, 2020. URL https://arxiv.org/abs/1812.07956.

Gabor Lugosi and Eulalia Nualart. Convergence of continuous-time stochastic gradient descent with applications to linear deep neural networks, 2024. URL https://arxiv.org/abs/2409.07401.

**Audience:**

Yes

**Audience Explanation:**

It's a paper on stochastic dynamics in neural networks, which is a core topic for TMLR's audience.

**Broader Impact Concerns:**

No concerns

**Claims And Evidence:**

Yes

**Claims Explanation:**

I didn't notice any major issues in the proofs. That said, I do not actively work in this area.

**Requested Changes:**

I would like to see a more detailed discussion of the technical challenges the authors encountered compared to prior work.

---

> ### Author Response · Authors · 2025-12-03
>
> We thank the reviewer for carefully reading our paper and for providing valuable and constructive feedback. Below, we address the questions and concerns raised. In the revised manuscript, all changes are highlighted in blue.
>
> **Comparison with prior work.** We agree that the distinctions between our contribution and existing results were not sufficiently clear for readers unfamiliar with the broader literature. To improve clarity, we added a short explanation in Section 1.1 and a more detailed discussion in Appendix A. The main difference from (Chizat et al., 2019) is that their analysis concerns deterministic gradient flow, i.e., a continuous–time model of full-batch gradient descent, while our work investigates a continuous–time approximation of stochastic gradient descent via stochastic differential equations (SDEs), whose convergence analysis in this work introduces substantial technical challenges which we address. Our results also differ from (Lugosi and Nualart, 2024), as we focus on the lazy-training regime, leading to weaker and more interpretable assumptions for overparameterized networks.
>
> We additionally note that a significantly revised version of (Lugosi and Nualart, 2024) appeared one week after our initial submission. This update includes results for nonlinear deep neural networks that were absent from the earlier version. We now discuss this in Appendix A.

---

### Review · Reviewer_nqeY · 2025-11-19

**Summary Of Contributions:**

This paper analyses stochastic gradient Langevin dynamics (SGLD) in the lazy training (also referred to as "kernel") regime, where the model parameters remain close to their random initialisation. Authors use an Itô SDE model of SGLD and, under some ~reasonable assumptions (regularity of loss and Hessian), they show that it can provide some convergence guarantees. In Th. 1 ("Convergence of SGLD) they show that the expected optimality gap decays exponentially fast (Eq. 3), while in Th. 2 they show that the probability that the training process leaves the lazy regime (i.e. it exits the sphere around the random initialisation point) can be made arbitrarily small by carefully selecting the scaling factor $\alpha$ (Eq. 9). Authors also provide specialisations of their results (Corollaries 1, "Risk convergence for shallow networks", and 2, "Risk convergence for deep networks"), where they show that the results in Th. 1 can be applied to deep networks under specific regularity conditions. Then authors experimentally validate their findings by simulating SGLD via Euler-Maruyama (a method for finding approximating solutions to SDEs).

**Additional Comments:**

I am not familiar with this domain -- I can't find anywhere in the TMLR review form to declare my level of expertise, hence I'm mentioning that here. I found this work interesting and novel (for me), but I am not deeply aware of the related literature on this topic. Please take this review with a grain of salt.

**Audience:**

Yes

**Audience Explanation:**

This is out of my domain of expertise but I am fairly sure many members of the community may find those findings interesting.

**Broader Impact Concerns:**

I do not think this is necessary

**Claims And Evidence:**

Yes

**Claims Explanation:**

Both theoretical findings and experimental results seem convincing.

**Requested Changes:**

- Can you please compare your assumptions to e.g. the ones Chizat et al.? Having a table showing the assumptions and corresponding findings in related works would help immensely for those not familiar with the related literature.

---

> ### Author Response · Authors · 2025-12-03
>
> We thank the reviewer for taking the time to read our paper and for providing valuable and constructive feedback. Below, we address the questions and concerns raised by the reviewer. In the revision, all modifications are highlighted in blue.
>
> **Comparison with prior work.** We agree that the distinctions between our contribution and prior work were not sufficiently clear to readers unfamiliar with the full literature. To address this, we added a brief clarification in Section 1.1 and a more extensive discussion in Appendix A. We also included a table summarizing the main differences between our work and prior work, in order to address the reviewer's suggestion.  The main difference between (Chizat et al., 2019) and our work is that they study deterministic gradient flow, which is a continuous-time model of (full-batch) gradient descent, whereas we study a continuous-time approximation of stochastic gradient descent via stochastic differential equations (SDE), which creates substantial technical challenges. Our work also differs from (Lugosi and Nualart, 2024) as we study the lazy training regime, which creates weaker and more interpretable assumptions for overparametrized networks.
>
> We would also like to note that a substantially updated version of (Lugosi and Nualart, 2024) was released one week after our initial submission. This new version includes results for nonlinear deep neural networks that were not present in the earlier version. We also discussed this in Appendix A in the revision.

---

### Decision · Action_Editor_8bFq · 2026-02-01

**Recommendation:** Accept as is

**Audience:**

Yes

**Audience Explanation:**

The lazy training regime of neural networks is an area of interest within the TMLR community.

**Claims And Evidence:**

Yes

**Claims Explanation:**

This paper presents a non-asymptotic convergence analysis of stochastic gradient Langevin dynamics in the lazy training regime. The authors establish exponential convergence in expectation to the empirical risk minimizer, provide explicit finite-time and finite-width bounds, and show that the neural tangent kernel remains non-degenerate with high probability under reasonable regularity assumptions. Theoretical results are supported by numerical experiments.

All reviewers agree that the work is an interesting contribution to the theoretical understanding of optimization dynamics in lazy regime of neural networks. While building on existing literature, the paper extends prior analyses in relevant directions, making it well suited for publication in TMLR.